# Reliability of a wearable wireless patch for continuous remote monitoring of vital signs in patients recovering from major surgery: a clinical validation study from the TRaCINg trial

Candice Downey,[1] Shu Ng,[2] David Jayne,[1] David Wong[3]

¹Leeds Institute of Medical Research at St. James's, University of Leeds, Leeds, UK
²School of Medicine, University of Leeds, Leeds, UK
³Centre for Health Informatics, University of Manchester, Manchester, UK

**Correspondence to**
Dr Candice Downey;
c.l.downey@leeds.ac.uk

## ABSTRACT

**Objective** To validate whether a wearable remote vital signs monitor could accurately measure heart rate (HR), respiratory rate (RR) and temperature in a postsurgical patient population at high risk of complications.

**Design** Manually recorded vital signs data were paired with vital signs data derived from the remote monitor set in patients participating in the Trial of Remote versus Continuous INtermittent monitoring (TRaCINg) study: a trial of continuous remote vital signs monitoring.

**Setting** St James's University Hospital, UK.

**Participants** 51 patients who had undergone major elective general surgery.

**Interventions** The intervention was the SensiumVitals monitoring system. This is a wireless patch worn on the patient's chest that measures HR, RR and temperature continuously. The reference standard was nurse-measured manually recorded vital signs.

**Primary and secondary outcome measures** The primary outcomes were the 95% limits of agreement between manually recorded and wearable patch vital sign recordings of HR, RR and temperature. The secondary outcomes were the percentage completeness of vital sign patch data for each vital sign.

**Results** 1135 nurse observations were available for analysis. There was no clinically meaningful bias in HR (1.85 bpm), but precision was poor (95% limits of agreement −23.92 to 20.22 bpm). Agreement was poor for RR (bias 2.93 breaths per minute, 95% limits of agreement −8.19 to 14.05 breaths per minute) and temperature (bias 0.82°C, 95% limits of agreement −1.13°C to 2.78°C). Vital sign patch data completeness was 72.8% for temperature, 59.2% for HR and 34.1% for RR. Distributions of RR in manually recorded measurements were clinically implausible.

**Conclusions** The continuous monitoring system did not reliably provide HR consistent with nurse measurements. The accuracy of RR and temperature was outside of acceptable limits. Limitations of the system could potentially be overcome through better signal processing. While acknowledging the time pressures placed on nursing staff, inaccuracies in the manually recorded data present an opportunity to increase awareness about the importance of manual observations, particularly with regard to methods of manual HR and RR measurements.

> **Strengths and limitations of this study**
>
> ► Surgical patients are a population likely to benefit from continuous physiological monitoring.
> ► A large number of paired data sets were available for comparison.
> ► The reference standard is a clinically relevant comparison, and is standard of care throughout the UK.
> ► The accuracy of the reference standard is user dependent.

## INTRODUCTION

Physiological monitoring using early warning score systems is effective but limited by its intermittent nature.[1] It is hypothesised that continuous vital signs monitoring may allow earlier detection of patient deterioration and thereby improve patient outcomes, but existing evidence is limited.[2] A consensus of international experts in safety and healthcare technology concluded that, if technically possible and affordable, all patients who are for active treatment should be continuously monitored.[3]

Until recently, continuous vital signs monitoring was limited to critical care areas because it required high staff-to-patient ratios and cumbersome equipment which tethered the patient to the bed-space, thereby inhibiting patient mobility and recovery. When hard-wired monitoring was implemented on a general ward, only 16% of patients remained connected in a 72 hour period.[4]

New remote monitoring devices, consisting of wearable sensors and aided by wireless data transmission, allow the patient to ambulate freely while enjoying the presumed advantages of extra monitoring. Since 2002, a number of such tools have received the US Food and Drug Administration clearance, indicating that they are safe and effective, but

BMJ

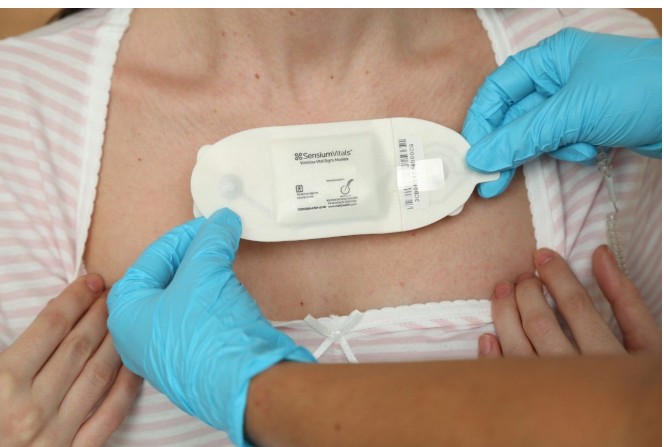

**Figure 1** The SensiumVitals monitoring patch. Image reproduced with permission from Sensium, Abingdon, UK.

clinical studies are required to demonstrate their utility in the inpatient setting.[5 6]

A remote monitoring device with a considerable amount of clinical evidence is the SensiumVitals patch (figure 1).[7–10] Attached to the patient's chest with two ECG electrodes, the device monitors heart rate (HR), respiratory rate (RR) and skin temperature continuously. The data are transmitted wirelessly every 2 min to a central monitoring station or a mobile device carried by the patient's nurse. This alerts the healthcare worker when there is deviation from preset physiological norms, alerting staff to potential patient deterioration.

The patch records RR by means of impedance pneumography and HR through single-lead ECG activity. Temperature is measured by a temperature-sensitive resistor. Once a physiological signal is fully acquired, it is processed by its associated embedded algorithm running inside the in-built processing unit, which enables the transmission of the resultant values to a nearby intranet hotspot for onwards transmission to the central monitoring system.

The underlying technology incorporated into such devices is well understood, but there is limited evidence for its reliability in the clinical setting. One previous study exists which validated the accuracy of the SensiumVitals system in 61 hospital patients. The patients were monitored at rest for a maximum of 2 hours, and the device was tested against a conventional bedside clinical monitor using capnographic RR.[11] This does not reflect the true clinical environment, which challenges such devices to provide monitoring continuity over several days in ambulatory patients.

In this study, we validated the accuracy of the SensiumVitals system to measure HR, RR and temperature in a postsurgical patient population at high risk for complications. The reference standard were manually recorded vital signs as part of the National Early Warning Score (NEWS). The objective of this study was to assess whether the wireless patch system is able to reliably measure vital signs continuously in the clinical setting, and to determine how well it compares to manually recorded measurements.

## METHODS
Informed consent to participate was obtained from all participants in the study.

### Study design
All participants were enrolled in the TRaCINg study, the protocol for which has been published previously.[12] This was a single-centre, feasibility, randomised, controlled, parallel group trial of continuous remote vital signs monitoring for patients who had undergone major elective general surgery at St James's University Hospital, Leeds, UK. Participants were individually randomised on a 1:1 basis to receive either remote monitoring plus NEWS or monitoring by NEWS alone. This paper describes the data from participants randomised to the remote monitoring arm, who wore the SensiumVitals patch during their hospital admission. The TRaCINg study is listed on the ISRCTN registry with study ID ISRCTN16601772 (http://www.isrctn.com/ISRCTN16601772).

### Patient and public involvement
Patients and the public were involved in the design of the randomised controlled trial, but were not involved in the design of this validation study.

### Data collection
Vital signs data were collected for each participant from two sources. The SensiumVitals vital sign data were documented at 2 min intervals and collected from a hospital desktop computer using data-acquisition software developed by Sensium. These data had been preprocessed to discard signals that were subject to gross electrical or motion artefact.[11] Patients were allowed to ambulate while wearing the monitoring patch; however, due to the major surgery they had undergone, most patients remained at their bedsides for the duration of their hospital stay.

NEWS data were collected at regular intervals, depending on the patients' status and based on the NEWS protocol.[13] Typically, vital signs were collected at the bedside, with the patients either sitting or lying down, by members of the nursing staff who were blinded to the SensiumVitals vital sign data: pulse rate was measured using the pulse oximeter on a multiparameter portable vital signs monitor; temperature was measured using a tympanic thermometer; RR was measured manually. The NEWS scores and their component parts were documented electronically. Researchers collected manually recorded HR, RR and temperature data from the hospital's electronic patient record. Other vital signs collected by the nursing staff as part of the early warning score, such as oxygen saturations, were not extracted.

### Data processing
The two data sources were linked using National Health Service (NHS) number and timestamp and consolidated

into a single deidentified spreadsheet. Paired data to a NEWS observation were derived from the SensiumVitals continuous data set by using the median vital sign value within a ±10 min window of a manually recorded observation. The time window was used to account for differences between the nurses' manually documented times and the automatic timestamps from the vital sign patch. The median value within this window was used to eliminate the impact of intermittent sensor noise.

## Outcomes

The primary outcomes were the 95% limits of agreement between manual nurse observations and wearable vital sign patch recordings of HR, RR and temperature (Temp). Following precedent, we defined clinical acceptability to be *max* ±10% for HR and RR (or ±3 breaths per minute or ±5 beats per minute) and 0.5°C for Temp.[14 15] The secondary outcome was the average percentage completeness of continuous patch data.

## Statistical analysis

For each vital sign, we first visually inspected the paired vital sign measurements via scatter plots, in addition to the raw time series vital signs from the Sensium patch.

Measurements were then formally compared using Bland-Altman analysis. In this analysis, the mean difference between the SensiumVitals data and the nurse observations and the 95% limits of agreement are calculated. We adjusted for multiple measurements from the same subject using a model in which time of measurement is modelled as a random effect.[16] This avoids bias caused by differences in number of measurements per patient. We also reported the Pearson correlation coefficient and the root mean squared (RMS) error for each vital sign.

In secondary analysis, we first assessed the average percentage completeness of the continuous patch data per patient. The numerator was defined as the number of 2 min periods in which vital sign data were provided by the patch. The denominator was the number of 2 min periods that span the time during which the patch was transmitting data. These time points were preferred to admission and discharge from ward times because the patch may not have been worn for the patient's entire ward admission. In sensitivity analyses, we repeated both the Bland-Altman analyses using ±2 and ±2 min windows of continuous data.

Analyses were undertaken using MATLAB R2017b and the R Methcomp package.[17 18]

## RESULTS

Fifty-one patients were recruited to the intervention arm of the TRaCINg study between October 2017 and April 2018. The median number of manually recorded observation sets was 19 per patient (range 2–73 sets of vital signs measurements). There were 1135 nurse observations available for analysis. All observations had a documented HR. Four observations had missing observations,

1 for RR and 3 for temperature. Vital sign traces for one participant over the course of their entire hospital stay are shown in figure 2.

## Heart rate

Figure 3 shows the scatterplot of nurse-recorded HR against the SensiumVitals patch. There is reasonable correlation between the two measurements ($R^2$=0.67, p<0.001). The mean and (SD) for manual and wearable HRs are 81.6 (16.2) beats per minute (bpm) and 84.3 (19.3) bpm, respectively. The mean percentage completeness of continuous patch data for HR was 59.2%. In addition, visual inspection of the example vital sign traces show good agreement between the measurements. The Bland-Altman bias (figure 4) was 1.85 bpm, with 95% limits of agreement −23.92 to 20.22 bpm. The RMS error was 11.25 bpm. The limits of agreement and RMS error exceeded the acceptability criterion.

## Respiratory rate

Figure 5 shows the scatterplot of nurse-recorded RR against the SensiumVitals patch data. There is no correlation between the two measurements methods ($R^2$=0.01, p<0.001). The mean and SD for manual and wearable RR were 17.6 (1.58) breaths per minute and 15.0 (5.5) breaths per minute, respectively. The mean percentage completeness of continuous patch data for RR data was 31.4%.

Visual inspection of the histogram for manually recorded RR shows a large peak at 18 breaths per minute, and a secondary peak at 16 breaths per minute. This result is unexpected for a natural physiological parameter, which may be expected to vary smoothly over the full range of values. Indeed, the peaks do not appear on the vital sign patch histogram. Inspection of the vital sign patch histogram indicates a significant proportion of measurements between 5 and 10 breaths per minute. No manually recorded RRs were recorded in this range. The Bland-Altman bias (figure 6) was 2.93 breaths per minute, with 95% limits of agreement −8.19 to 14.05 breaths per minute. The RMS error was 6.14 breaths per minute and the limits of agreement are wider than the pre-specified acceptable error of 3 breaths/min.

## Temperature

Figure 7 is a scatterplot of temperatures recorded by nurses vs those measured by the SensiumVitals patch. Histograms for each measurement method are presented alongside the x-axis and y-axis. There is low correlation between the two measurement methods ($R^2$=0.13, p<0.001). The mean and (SD) of manual temperature and wearable temperature were 37.1°C (0.5°C) and 36.4°C (1.0°C). Further inspection of the vital sign time series in figures 1 and 2 shows multiple clinically implausible fluctuations of up to 2°C within 2 hours within each time series. The mean percentage completeness of continuous patch data for temperature was 72.8%.

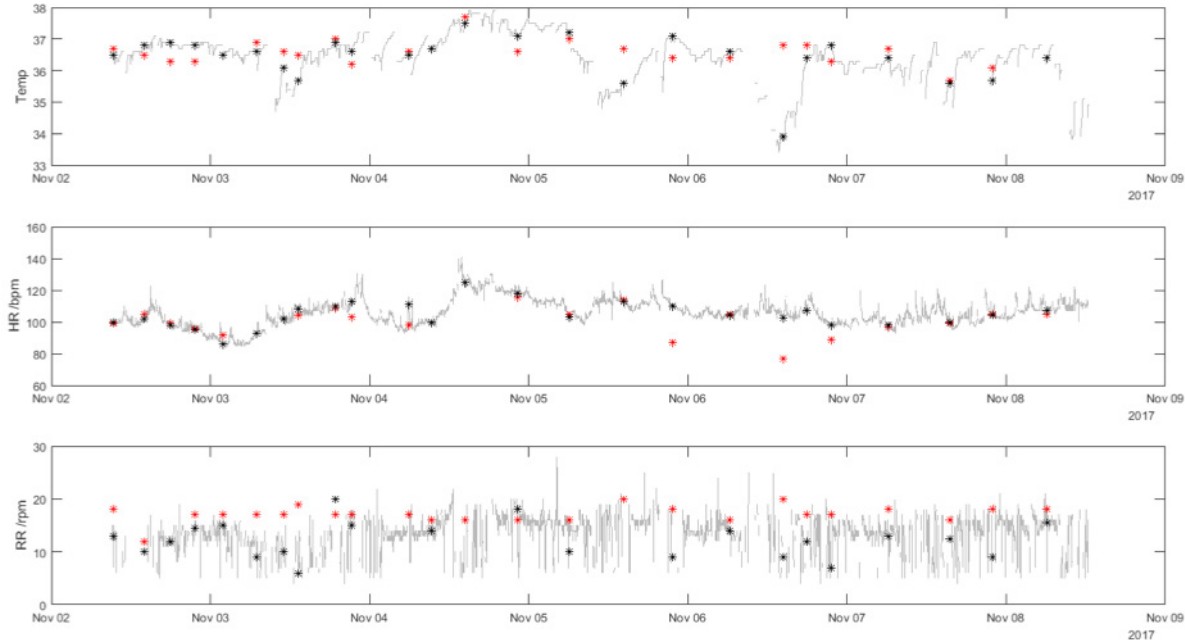

**Figure 2** Vital signs data for a single participant. The grey lines show the minute-by-minute vital sign values from the SensiumVitals patch. The black markers show the median value of the SensiumVitals vital signs (evaluated from ±10 mins of the nurse observation time). The red markers show the manually recorded vital signs. Where there is a wide difference between the red and black markers at a single time point, this indicates disagreement between the two vital signs measurement techniques. HR, heart rate; RR, respiratory rate; Temp, temperature.

Initial visual inspection was therefore sufficient to show that the patch-derived temperature is not a suitable proxy for core temperature, as measured by tympanic thermometer. The Bland-Altman bias (figure 8) was 0.82 ℃, with

95% limits of agreement −1.13 ℃ to 2.78 ℃. The RMS error was 1.28 ℃. In addition to large systematic bias between the two methods, the limits of agreement did not meet the predefined clinical acceptability criterion (0.5°C).

In a sensitivity analysis, all Bland-Altman analyses was repeated using ±2 and ±5 min windows of vital sign patch data. There were no meaningful differences in the bias or limits of agreement (Online supplementary material).

## DISCUSSION

In this 51 patient validation study, temperature, RR and HR measurements obtained from a wearable vital sign patch were compared with manually recorded

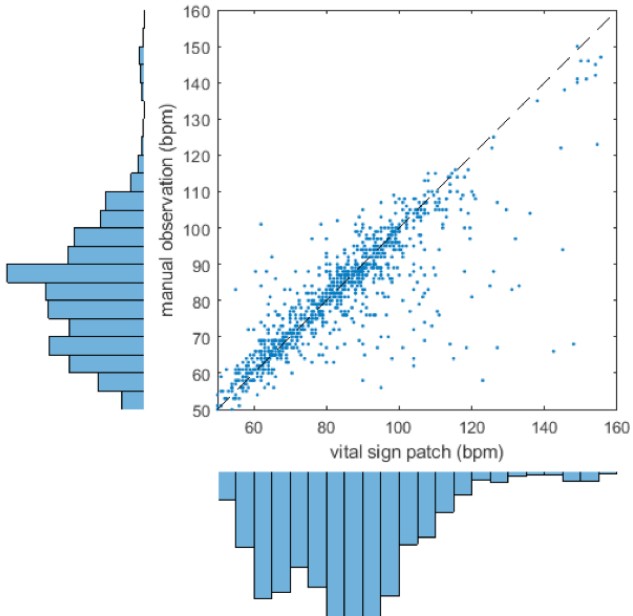

**Figure 3** Scatter plot and marginal histogram of paired manual and SensiumVitals heart rate observations. bpm, beats per minute.

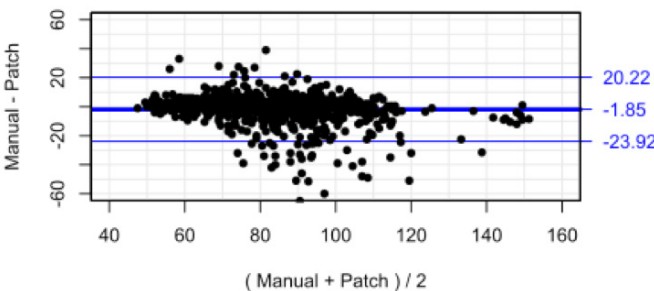

**Figure 4** Bland-Altman plot for heart rate with limits of agreement adjusted for repeated measures.

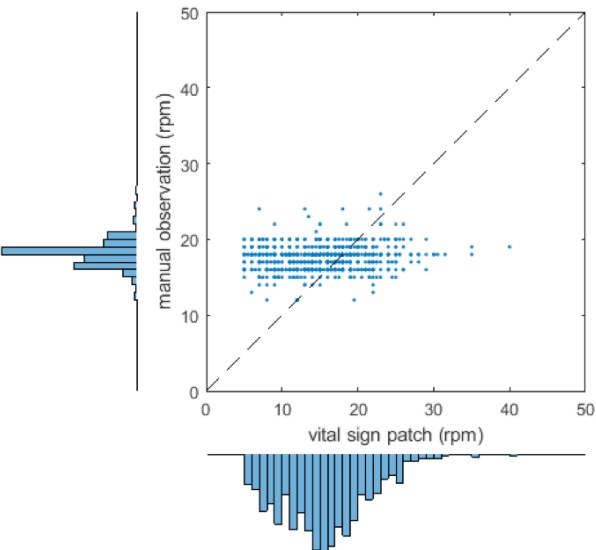

**Figure 5** Scatter plot and marginal histogram of paired manual and SensiumVitals respiratory rate observations. rpm, respirations per minute.

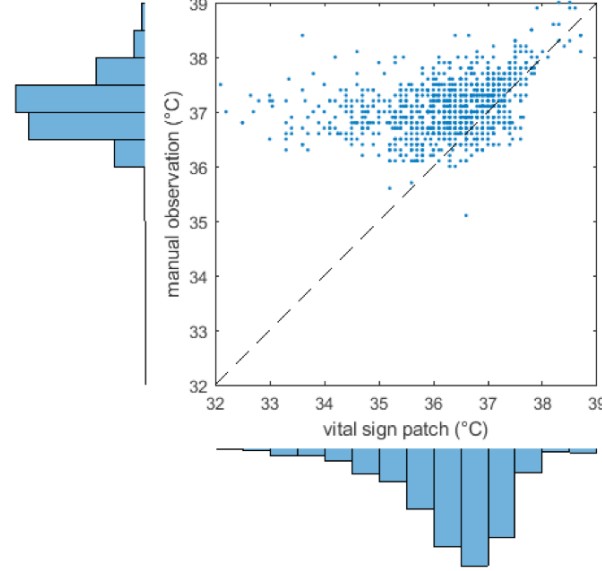

**Figure 7** Scatter plot and marginal histogram of paired manual and SensiumVitals temperature observations.

observations by nursing staff. While there was reasonable correlation between the two methods for HR measurements, there were large discrepancies in many instances, as indicated by the Bland-Altman analysis. It is not clear whether there were errors in the manual observation, in the vital sign patch, or both. There was low correlation for RR and temperature. The differences between manual and vital sign patch measurements for all three measured vital signs were outside of acceptable limits.

An advantage of the study design is the collection of a large number of data points for analysis. The approach is clinically valid, as the NEWS system is the national standard for vital signs monitoring in the UK. The surgical patient population is a clinically relevant cohort. There are high rates of complications after major surgery,[19] but many surgical complications, such as sepsis, are attenuated by early detection. By virtue of their suitability for surgery, patients experiencing severe complications are likely to be candidates for full active management and escalation of care. They are therefore a population

likely to benefit from reliable continuous physiological monitoring.

There are few clinical evaluations of continuous vital signs monitoring in the literature. Previous validation studies have studied participants who are confined to their bed space by wired monitoring equipment.[11 14] In the surgical setting, enhanced recovery programmes mandate early mobilisation after surgery. In this study, patients were allowed to ambulate freely as part of their usual postoperative care, which may have produced some motion artefact on the continuous monitoring data; this may explain why the findings from this study show worse correlation when compared with previous studies which compared two stationary measurements. The patch algorithms are designed to identify and reject physiological signals corrupted by significant sources of noise inherent to the ambulatory nature of wireless monitoring; however, it is possible that RR data may have shown artefact from speech.

The findings must be interpreted within the limitations of the study. There were a relatively small number

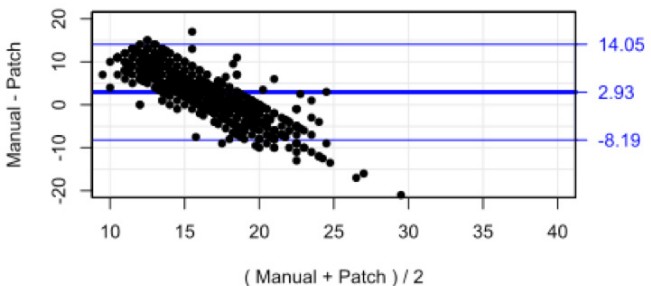

**Figure 6** Bland-Altman plot for respiratory rate with limits of agreement adjusted for repeated measures.

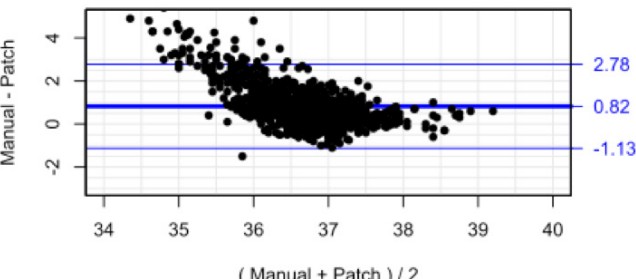

**Figure 8** Bland-Altman plot for temperature with limits of agreement adjusted for repeated measures.

of patients in the study. Data completeness from the vital sign patch was low, especially for RR, although results for HR and temperature were similar to previous work.[14] The reference standard, while clinically relevant, is inherently flawed. Early warning scores such as NEWS are known to be limited by their user-dependent nature. Time and staffing pressures placed on nursing staff in an increasingly busy clinical environment may be driving the adoption of time-saving, less accurate techniques; in this study, HR was typically inferred from the pulse rate measured by a pulse oximeter, despite the fact that this is known to be less accurate than manual palpation of the radial pulse. In addition, manually collected vital signs can be subject to the effects of 'white-coat hypertension'; HR, RR and temperature can be elevated simply by the arousal effect of the nurse interaction.[20]

Deficits in the manually recorded observations were particularly evident in the analysis of RR. Analysis of the manually recorded values alone revealed a statistically unlikely preponderance of 18 breaths per minute, with a secondary peak at 16 breaths per minute. These peaks were not visible for the vital sign patch, suggesting that this is a measurement artefact in the way that manual measurements are made, rather than a real effect. It has been well described that RR is often miscalculated or omitted when calculated early warning scores.[21 22] It is also recognised that clinical staff detect patient status in advance of manual measurements for an early warning score system 'by using information not currently encoded within it'.

The patch data for RR are also unlikely to be reliable, as a significant proportion of measurements were between 5 and 10 breaths per minute. This proportion of low values is much greater than those described in previously derived distributions from larger populations.[23] There are also rapid fluctuations in RR which are physiologically implausible and may have been affected by patient movement, speech or coughing.

The manually recorded temperature measurements showed plausible distributions and are likely to be accurate. The high bias between the nurse-measured temperatures and the patch data can be explained by the difference in measurement techniques. The patch measures skin temperature which may not accurately reflect the tympanic temperature measured by the nursing staff. Skin temperature is highly dependent on environmental factors such as the ambient temperature, clothing and blankets.

The reliability of the continuous temperature measurement is, however, limited. The time series analysis shows evidence of regular patch disconnection, indicated by rapid drops in temperature followed by increases consistent with conductive heating, or warming back up. These warm-up periods render the raw signals unreliable, although this limitation may be overcome through better signal processing. For instance, Clifton *et al* used Bayesian change point analysis to detect step changes in temperature across a large study population. A similar approach

may be used to determine disconnection on an individual patient basis.[24]

## CONCLUSIONS

The differences between manual and vital sign patch measurements for all three measured vital signs were outside of acceptable limits. On some occasions, this may be due to artefact in the continuous signal; this could be overcome through better signal processing. Other discrepancies may be due to errors during manual measurement. While acknowledging the time pressures placed on nursing staff, inaccuracies in the manually recorded data present an opportunity to increase awareness about the importance of manual observations, particularly with regard to methods of manual HR and RR measurements.

**Contributors** CD, SN and DW were involved in the conception of the work. CD designed the study. DW provided methodological expertise. SN undertook the data collection. DW and CD performed the analysis and interpretation. CD and DW drafted the article. CD, SN, DW and DJ were involved in critical revision of the article and have given final approval of the version to be submitted.

**Funding** Candice Downey is in possession of a Doctoral Research Fellowship (DRF-2016-09-037) supported by the National Institute for Health Research. DGJ received funding support through an NIHR Research Professorship. The research is supported by the NIHR infrastructure at Leeds.

**Disclaimer** The views expressed in this publication are those of the authors and not necessarily those of the NHS, the National Institute for Health Research, Health Education England or the Department of Health.

**Competing interests** None declared.

**Patient consent for publication** Not required.

**Ethics approval** Ethical approval was granted on 10 October 2017 by the Yorkshire & The Humber—Leeds West Research Ethics Committee, ref: 17/YH/0180.

**Provenance and peer review** Not commissioned; externally peer reviewed.

**Data availability statement** Data are available upon reasonable request.

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
