## [Reviewer comments · BMJ Open]

ARTICLE DETAILS

TITLE (PROVISIONAL)	Reliability of a wearable wireless patch for continuous remote monitoring of vital signs in patients recovering from major surgery: a clinical validation study from the TRaCINg trial
AUTHORS	Downey, Candice; Ng, Shu; Jayne, David; Wong, David

VERSION 1 - REVIEW

REVIEWER	Jarkko Harju Tampere University Hospital
REVIEW RETURNED	17-May-2019

GENERAL COMMENTS	Dear Editor I read the manuscript from C. Downey et al with great interest. Remote monitoring is increasingly studied and developed and is likely to have a major role in patient monitoring. There are several devices being studied but the evidence in real clinical situations is still limited. The manuscript is structured and the language is generally fluent and easy to read. However, there are some major concerns regarding the article which are pointed below. Abstract: 1. results: The primary and secondary outcomes indicate the limits of agreement as the main results. However, those are not reported in the results section but instead there are correlations. The correlation between two measurements is not appropriate to describe the similarity or difference between two methods. Instead the Bland Altman numbers should be reported as the main result.2. conclusions: What is meant with heart rate capable of reliably measuring heart rate? The accuracy (mean difference) was rather good (-1,85 ppm which can be considered acceptable. However, the precision (LoA -23.92 to 20.22) is rather low. I do disagree that the heart rate measurements were reliable with this wide limits of agreement. Introduction 3. Page 5 Line29: The references are missing for “considerable amount of evidence” mentioned in the text. Please add these also at this point. Some of them are discussed elsewhere in the text.4. Page 5 line 50: The reference 7 is also a clinical study but the patients were stationary. In this current study are the patients stationary during the manual measurement? Methods section – 2.3. data collection and 2.4 Data processing:
---

5. The electronic data was integrated into manually collected data using a time stamp from patient files and a 20 minutes (?) overall window for measuring the median value from patch. Does this mean that the manual recordings were point measurements when patient must have been stationary to be able to measure the stats and SensiumVital data has actually been measured within 20 minutes of the manual measurement? Have the patients been also been stationary while measuring used SensiumVital median values? If not, are the measurements comparable using this wide window? Were the results similar if a more narrow window for time stamps was measured? If the SensiumVital measurements were taken during movement it is likely that the measurement pairs are not comparable which might explain the wide limits of agreement.

Statistical analysis:

6. How is the Bland Altman data calculated? Please ad the reference of the method used. Have authors used the Bland Altman for multiple methods per subject.

7. The authors could ad the root-mean-square error to describe the difference between the devices.

2.5. Outcomes:

8. Did the results meet these definitions for clinical acceptability? These are not commented in results section. Also, the order of the measurements differs in abstract, outcomes and results section. Using consistently the same order in reporting values would help in reading (here HR, RR, Temperature)

Results:

9. Is the 2737 pairs of data valid for all measurements or are there different number of pairs for HR, RR and temperature measurements?

10. Likely the mean percentage for completeness means readings available for SensiumVital data and not for data pairs? Please type more clearly at each point.

11. Page 8, lines 24-: Was the large peak mentioned in histogram at 18 and 16 peak/minute from same patients since there were wide range of measurement pairs per patient ranging from 2 to 73 sets per patient. Having measurements from same patient at same kind of conditions might increase the likelihood for same measurement reading.

12. What type of correlation was used (Spearman, Pearson)? Please mention in statistical section and ad p-values.

Discussion:

13. The correlation is not sufficient to describe the difference between two methods. Use instead the Bland Altman Plot values and interpret accordingly.

14. Page 9 Line 13-: I do disagree with this paragraph. The previous studies (references 7 and 10) have used two continuous measurements with good time stamps to link the data. The stationary of the patients limits the generalizability of the results if the patients are moving which will increase the motion artefact. In the current study the patients were allowed to move freely, but the used comparison is stationary measurement. It is true that the data completeness may be limited because the patients are able to move. In the current study the accuracy difference can not be shown but also the comparative method should be portable (such as Holter device).

	15. The previous studies have shown better results for the measurement. Authors could speculate why the findings of current manuscript are worse. 16. page 9 line 39-: The speech will affect the respiratory rate measurement and as though the measurement is only valid when the patient is breathing freely. Were these kind of measurement periods with imminent artefact ruled out from the comparison? 17. Page 9 line 47-: The reference 11 mentioned in the methods section states that also tympanic measurement is likely to be inaccurate when measuring core temperature. The low precision of the comparing measurement may also affect the reliability of the data comparison. Conclusion: 18. Page 10 lines 9-: I do disagree with the conclusion as mentioned above. The correlation is not sufficient statistic to determine the difference between two methods.
--	---

REVIEWER	Professor Julie Considine Deakin University - Eastern Health, Victoria, Australia
REVIEW RETURNED	17-May-2019

GENERAL COMMENTS	Thank you for submitting your work to BMJ Open. This is an important study and a well written paper. I had only minor comments (please see attached file) but I think the conclusion regarding lack of awareness of this too simplistic and does not reflect the complexity of the issues examined in this study. The reviewer provided a marked copy with additional comments. Please contact the publisher for full details.
--

VERSION 1 – AUTHOR RESPONSE

Reviewer: 1

Abstract:

1. Results: The primary and secondary outcomes indicate the limits of agreement as the main results. However, those are not reported in the results section but instead there are correlations. The correlation between two measurements is not appropriate to describe the similarity or difference between two methods. Instead the Bland Altman numbers should be reported as the main result.

Thank you for this observation. We have added the Bland Altman results to the abstract.

2. Conclusions: What is meant with heart rate capable of reliably measuring heart rate? The accuracy (mean difference) was rather good (-1,85 ppm which can be considered acceptable. However, the precision (LoA -23.92 to 20.22) is rather low. I do disagree that the heart rate measurements were reliable with this wide limits of agreement.

We have altered the text in this section to reflect this pertinent observation: 'The remote continuous monitoring system correlates well with manually-recorded heart rate, although wide limits of agreement indicate a low level of precision.'

Introduction:

3. Page 5 Line29: The references are missing for "considerable amount of evidence" mentioned in the text. Please add these also at this point. Some of them are discussed elsewhere in the text.

We have added three exemplar references at this point in the text.

4. Page 5 line 50: The reference 7 is also a clinical study but the patients were stationary. In this current study are the patients stationary during the manual measurement?

Typically, the patients were sitting or lying down during their manual measurements. We have added this information for clarity in to Section 2.3 (Data Collection). We have also included more information about patient ambulation during continuous monitoring in the same section.

Methods section – 2.3. data collection and 2.4 Data processing:

5. The electronic data was integrated into manually collected data using a time stamp from patient files and a 20 minutes (?) overall window for measuring the median value from patch. Does this mean that the manual recordings were point measurements when patient must have been stationary to be able to measure the stats and SensiumVital data has actually been measured within 20 minutes of the manual measurement? Have the patients been also been stationary while measuring used SensiumVital median values? If not, are the measurements comparable using this wide window? Were the results similar if a more narrow window for time stamps was measured? If the SensiumVital measurements were taken during movement it is likely that the measurement pairs are not comparable which might explain the wide limits of agreement.

In response to these astute comments, we performed a sensitivity analysis for heart rate; the Bland-Altman analysis was repeated using ± 2 and ± 5 minute windows of vital sign patch data. There were no meaningful difference in the bias or limits of agreement. We have provided this data as Supplementary Material.

Statistical analysis:

6. How is the Bland Altman data calculated? Please add the reference of the method used. Have authors used the Bland Altman for multiple methods per subject.

This information has been added to the Statistical Analysis section.

7. The authors could add the root-mean-square error to describe the difference between the devices.

Thank you for this suggestion. We have performed this analysis and added it to the manuscript.

2.5. Outcomes:

8. Did the results meet these definitions for clinical acceptability? These are not commented in results section. Also, the order of the measurements differs in abstract, outcomes and results section. Using consistently the same order in reporting values would help in reading (here HR, RR, Temperature)

We have added a commentary regarding whether the results met the definitions for clinical acceptability in the Results section. We have reordered the Results section to reflect the same order as the Abstract and Methods sections, and relabelled the figures accordingly.

Results:

9. Is the 2737 pairs of data valid for all measurements or are there different number of pairs for HR, RR and temperature measurements?

All observations had a documented heart rate. Four observations had missing observations, 1 for respiratory rate and 3 for temperature. We have added this information to the Results section.

10. Likely the mean percentage for completeness means readings available for SensiumVital data and not for data pairs? Please type more clearly at each point.

In Section 2.5 (Outcomes) we have clarified the definition of this term: 'defined as the percentage of available data from the patch for every point of manually-recorded data.'

11. Page 8, lines 24-: Was the large peak mentioned in histogram at 18 and 16 peak/minute from same patients since there were wide range of measurement pairs per patient ranging from 2 to 73 sets per patient. Having measurements from same patient at same kind of conditions might increase the likelihood for same measurement reading.

It is unlikely that the peaks were from the same patients, as these peaks were not seen in the patch data. Assuming that the patch is not just generating random numbers, then the absence of peaks in the patch data means that the 16rpm and 18 rpm peaks are not likely to be accurate values. To be absolutely sure, we have produced a histogram for respiratory rate, weighted by patient (so that observations for patients with a longer stay contribute less), and you can see that the peaks are still present.

12. What type of correlation was used (Spearman, Pearson)? Please mention in statistical section and ad p-values.

The type of correlation has been added to the Statistical Analysis section: 'We also reported the Pearson correlation coefficient and the root mean squared (RMS) error for each vital sign.' P-values have been added to the results.

Discussion:

13. The correlation is not sufficient to describe the difference between two methods. Use instead the Bland Altman Plot values and interpret accordingly.

We have added the raw values to the Discussion and provided an interpretation based on the clinical acceptability thresholds defined in the Methods section.

14. Page 9 Line 13:- I do disagree with this paragraph. The previous studies (references 7 and 10) have used two continuous measurements with good time stamps to link the data. The stationary of the patients limits the generalizability of the results if the patients are moving which will increase the motion artefact. In the current study the patients were allowed to move freely, but the used comparison is stationary measurement. It is true that the data completeness may be limited because the patients are able to move. In the current study the accuracy difference cannot be shown but also the comparative method should be portable (such as Holter device).

We have deleted the statement that previous methodologies lack validity in the surgical population. We have added a commentary explaining that the difference between ambulatory and stationary monitoring may explain the disparity between this and existing studies.

15. The previous studies have shown better results for the measurement. Authors could speculate why the findings of current manuscript are worse.

We have added a sentence to this effect the Discussion.

16. page 9 line 39:- The speech will affect the respiratory rate measurement and as though the measurement is only valid when the patient is breathing freely. Were these kind of measurement periods with imminent artefact ruled out from the comparison?

The patch algorithms are designed to identify and reject physiological signals corrupted by significant sources of noise inherent to the ambulatory nature of wireless monitoring. I have added a short commentary on this in the Discussion.

17. Page 9 line 47:- The reference 11 mentioned in the methods section states that also tympanic measurement is likely to be inaccurate when measuring core temperature. The low precision of the comparing measurement may also affect the reliability of the data comparison.

The difference here is between skin temperature and tympanic temperature; unfortunately neither of the two monitoring methods measure core temperature.

Conclusion:

18. Page 10 lines 9:- I do disagree with the conclusion as mentioned above. The correlation is not sufficient statistic to determine the difference between two methods.

We have removed the mention of correlation from the Conclusions section.

Reviewer: 2

The conclusion regarding lack of awareness of this too simplistic and does not reflect the complexity of the issues examined in this study.

We have added a short commentary about this in the Discussion, and amended the final sentence in both the Abstract and the Conclusions sections, for clarity.

Page 3 (abstract, conclusions): Do you think inaccuracies in manually-recorded vital signs is due to lack of awareness. I think all RNs are aware of the importance of vital sign assessment but there are significant system and workload issues that may be plausible explanations for inaccuracies? for me, this statement is too simplistic in the face of no evidence to suggest staff are unaware.

We agree. Inaccuracies can certainly be due to the time pressures placed on staff, and this may well explain time-saving measures such as extrapolating respiratory rates from previous recordings and measuring pulse rates using the pulse oximeter. However, in our experience, there are some knowledge gaps which could be easily remedied. There is certainly a lack of awareness in our institution that pulse rate from the pulse oximeter is not as accurate as manual palpation of the radial pulse, especially when an arrhythmia is present. Increasing awareness of the importance of manual pulse measurements may go some way towards improving the accuracy of vital signs monitoring in this instance. We have added a short commentary about this in the Discussion, and amended the final sentence in both the Abstract and the Conclusions sections, for clarity.

Page 5 (Introduction): is validated the right word given it implies positive relationships?

We have used the word 'validate' to provide consistency with the nomenclature of other studies with similar methodologies.

Page 6 (methods, Data Collection): so is therefore, by definition, also intermittent and not continuous?

The patch device monitors vital signs continuously. The vital signs data is documented every 2 minutes after smoothing of the data points by the in-built algorithm, as described in Paragraph 4 of the Introduction.

Page 6 (Methods, Data Collection): using pulse oximetry to measure pulse rate is flawed and would not be in line with most vital sign standards in most countries that would state that pulse rate should be measured by palpation of the radial pulse.

We agree. Measuring pulse rate with pulse oximetry is not compliant with local or national guidelines. However, unfortunately it is typically the way that 'heart rate' is measured on the study wards, despite the fact that manual palpation is more accurate. I have added a short commentary on this in the Discussion.

Page 9 (Discussion, Paragraph 2): if continuous monitoring devices are reliable

We have added the word 'reliable' to this sentence for clarity.

Page 9 (Discussion, Paragraph 4): needs supporting reference.

We have added a supporting reference here.

Page 10 (Conclusions): see same comment as abstract - this is a very simplistic statement that underplays the complexities of clinical care.

We have added a short commentary about this in the Discussion, and amended the final sentence in both the Abstract and the Conclusions sections, for clarity.

VERSION 2 – REVIEW

REVIEWER	Jarkko Harju Tampere University Hospital
REVIEW RETURNED	02-Jul-2019

GENERAL COMMENTS	Dear Editor Thank you for the opportunity to make the revision for this second version of the manuscript. The manuscript has improved and have only minor comments to add. Page 8 line 23: The authors added the commentary on Bland-Altman plot. This chapter contains mostly results and not commentary and should therefore be places in results section.
--

VERSION 2 – AUTHOR RESPONSE

Reviewer: 1

Page 8 line 23: The authors added the commentary on Bland-Altman plot. This chapter contains mostly results and not commentary and should therefore be placed in results section.

We have deleted this repetition of the results from the Discussion section, as we agree that it is unnecessary.